# Prediction of true *Helicobacter pylori-*uninfected status using a combination of age, serum antibody and pepsinogen: Logistic regression analysis

**Takako Takayama[1], Hideo Suzuki[1,2]\*, Kosuke Okada[1], Takeshi Yamada[2], Kazushi Maruo[3], Yoko Saito[4], Yuji Mizokami[2]**

**1** Tsukuba Preventive Medical Research Center, University of Tsukuba Hospital, Tsukuba, Japan, **2** The Department of Gastroenterology, Faculty of Medicine, University of Tsukuba, Tsukuba, Japan, **3** Department of Biostatistics, Faculty of Medicine, University of Tsukuba, Tsukuba, Japan, **4** Ibarakiken Medical Center, Mito, Japan

\* hideoszk@md.tsukuba.ac.jp

**Data Availability Statement:** All relevant data are within the manuscript.

## Abstract

### Introduction

To prevent gastric cancer, it is important to accurately determine the presence of *Helicobacter pylori* (HP) infection. However, correctly identifying HP-uninfected individuals is difficult when using the combination of HP antibody and pepsinogen (PG).

### Objective

The aim of this study was to discriminate true HP-uninfected individuals from others without the need for endoscopic examination.

### Methods

A total of 684 subjects with no history of HP eradication who underwent a medical checkup at our hospital were enrolled. The "true uninfected individuals" were determined by a negative stool antigen test and no endoscopic findings of HP-associated gastritis. HP antibody was measured by the latex immunoassay method. Logistic regression analysis using a combination of noninvasive parameters was performed to develop a formula for predicting true uninfected individuals.

### Results

A total of 528 subjects were classified as true uninfected individuals. Logistic regression analysis showed that statistically significant factors for true uninfected individuals were age ($p < 0.001$), HP antibody ($p < 0.001$), PGI ($p < 0.001$), and PGII ($p = 0.012$). The areas under the curve (AUCs) for true uninfected individuals were the highest (0.944) upon applying the prediction formula including four parameters: age, HP antibody, PGI, and PGII. Both the sensitivity and the specificity of the four-parameter prediction formula were higher than

**Funding:** The author(s) received no specific funding for this work.

**Competing interests:** The authors have declared that no competing interests exist.

those of the traditional three-parameter model using HP antibody, PGI, and PGI/II ratio (sensitivity: 93.2% vs. 86.6% and specificity: 88.5% vs. 82.7%).

## Conclusions

Our findings suggest that a model with a combination of four noninvasive parameters is useful for predicting true HP-uninfected individuals without the need for endoscopic examination.

## Introduction

The ABC classification, which is based on the *Helicobacter pylori* (HP) antibody titer determined by enzyme-linked immunosorbent assay (ELISA) and the levels of serum pepsinogens (PGs), is useful for discriminating populations at high and low risk of developing gastric cancer [1, 2]. In this method, the subjects are classified into four groups: Group A [HP antibody (–), PG(–)], Group B [HP antibody(+), PG(–)], Group C [(HP antibody(+), PG(+)], and Group D [HP antibody(–), PG (+)]. The risk of gastric cancer is reported to be increased in Groups B, C, and D compared with that in Group A [3]. For individuals in Groups B, C, and D, endoscopy is recommended. However, the ABC classification is considered to have the limitation that some subjects with current and past infections are included in Group A, which is considered the low-risk group (false A).

Chinda et al. showed that approximately 20% of individuals in Group A had current or past HP infection [4]. In addition, Kotachi et al. reported that 44.4% of patients in Group A showed atrophic gastritis (C-2 or higher). Moreover, even after they excluded those with negative results for high titers (3–9.9 U/ml) of HP antibody, they found that 36.4% of those in Group A still had atrophic gastritis [5]. HP gradually disappears during the deterioration of atrophic gastritis [6, 7]. In addition, HP is incidentally eradicated in some cases because of the administration of antibiotics that are commonly used for other diseases. As a result, cases with some risk of gastric cancer are misclassified into Group A. Meanwhile, those who are free from HP infection are thought to be at very low risk for gastric cancer. Indeed, a prospective study in Japan showed that, over 10 years, gastric cancer was estimated to occur in 5% of infected patients and no uninfected patients [8].

Previous studies effectively identified the cut-off level of pepsinogen for predicting cases with the incidental eradication of HP infection in Group A [4, 9]. However, to the best of our knowledge, few reports have been published on studies in which true uninfected status itself was directly diagnosed using HP antibody levels, PGs, and several other factors.

In addition, for HP antibody detection, the latex immunoassay (LIA) method has recently begun to replace the ELISA method, but the optimal cut-off value for identifying true HP-uninfected individual using LIA method remains unclear.

Prediction models are efficient in correctly distinguishing non-diseased from diseased individuals without the need for invasive procedures [10, 11]. The present study aimed to develop an accurate method for predicting true HP-uninfected individuals using logistic regression analysis with noninvasive parameters without the need for endoscopic examination.

## Materials and methods

A total of 1,519 patients who visited Tsukuba Preventive Medicine Research Center, University of Tsukuba Hospital, from April 2017 to April 2019, were analyzed. Serum HP antibody, PG, and stool antigen test (SAT) evaluations were requested for all patients. We excluded 467 patients who had a history of HP eradication therapy, 250 patients who did not undergo endoscopy, 15 patients who did not undergo SAT, 7 patients who did not undergo PG testing, 5 patients who had a history of gastrectomy, 1 patient with renal dysfunction (serum creatinine level ≥3 mg/dL), 79 patients who were taking proton pump inhibitors, and 2 patients with implausible results (PGI ≥1000 ng/ml) (Fig 1).

HP antibody level was measured by the LIA method (Eiken Chemical Co., Ltd., Tokyo, Japan). The SAT was performed using the Testmate Pylori Antigen EIA (Wakamoto Pharmaceutical, Tokyo, Japan). PGs were measured by CLIA (LSI Medience, Tokyo, Japan). A gastroenterologist performed the endoscopic examination and diagnosed the presence or absence of atrophic gastritis based on the Kimura-Takemoto classification [12]. We defined HP-associated gastritis by the findings of atrophy (C-2 over) or metaplasia in accordance with the Kyoto Classification [13], which was confirmed by two gastroenterologists.

By using endoscopic findings and SAT, we categorized patients into three groups: true uninfected [HP-associated gastritis(–) and SAT(–)], currently infected [HP-associated gastritis(+) and SAT(+)], and spontaneously eradicated cases [HP-associated gastritis(+) and SAT(–)]. We excluded nine patients [HP-associated gastritis(–) and SAT(+)] because their infection

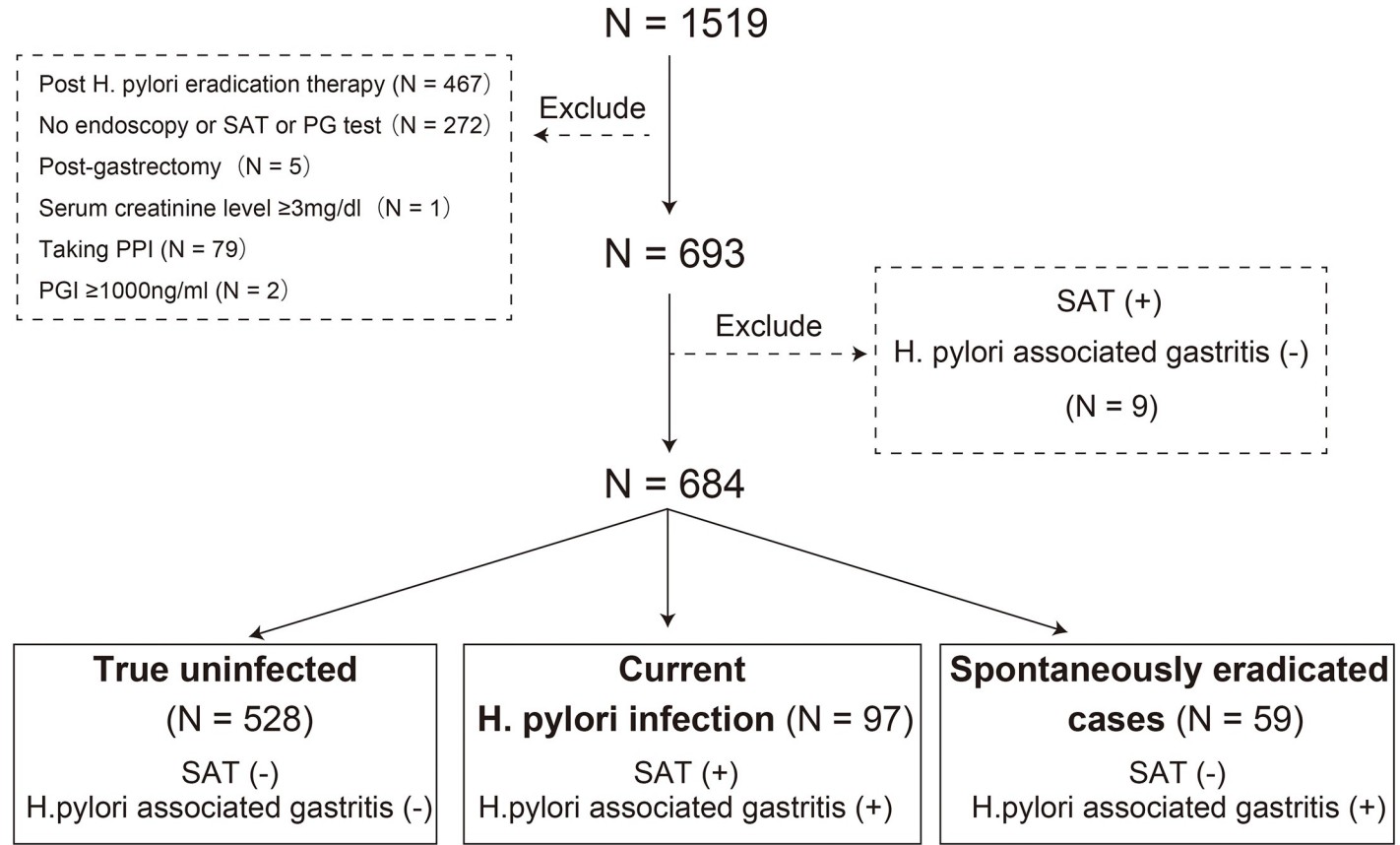

**Fig 1. Flowchart of the study.**

status was difficult to identify. Finally, 684 patients (528 true uninfected, 97 current infected, and 59 spontaneously eradicated cases) were enrolled for our observational study (Fig 1).

Baseline data for noninvasive parameters, including sex, age, HP antibody, PGI, PGII, and PGI/II ratio, in the three groups were compared by the Mann-Whitney U test. Using these six parameters, we performed binomial logistic regression analysis to develop a formula for predicting true uninfected individuals. Next, we used the chi-squared test [14] to compare the areas under the curve (AUCs) among models with A) HP antibody alone; B) HP antibody, PGI, and PGI/II ratio; C) age, HP antibody, PGI, and PGII; and D) age, sex, HP antibody, PGI, PGII, and PGI/II ratio. We defined the point on the ROC (receiver operating characteristic) curve with the minimum distance from the upper left corner (0, 1) as the optimal cut-off value. Finally, we compared the diagnostic accuracy between our best prediction formula and the three-parameter model (HP antibody <3 U/ml or <10 U/ml and PGI >70 ng/ml or PGI/II ratio >3.0) for identifying HP-uninfected individuals. All statistical analyses were performed using Bell Curve for Excel version 3.20 (Social Survey Research Information Co., Ltd., Tokyo, Japan); $p$ <0.05 was considered statistically significant. HP antibody <3.0 U/ml was calculated as 1.5 U/ml. The study was approved by the University Hospital of Tsukuba Ethics Committee (H29-304). All data were fully anonymized before we accessed them and informed consent was obtained by opt-out method under approval from the ethics committee.

## Results

The medians (IQRs) in true uninfected, currently infected, and patients with spontaneously eradicated infection were as follows: age: 54.0 (44.0–65.0), 64.0 (55.0–70.0), 67.0 (59.0–71.5); antibody titer: 1.5 (1.5–1.5), 29.0 (14.0–100.0), 3.0 (1.5–9.5); PGI: 47.9 (40.0–59.0), 68.3 (52.7–84.2), 37.5 (26.0–52.9); PGII: 6.7 (5.3–8.5), 21.5 (15.6–27.1), 6.4 (5.4–7.9); and PGI/II ratio: 7.3 (6.4–8.3), 3.3 (2.4–3.9), 5.3 (4.0–7.2), respectively (Table 1). The factors that were statistically significant among all three groups were HP antibody, PGI, and PGI/II ratio. Age was significantly lower in the true uninfected individuals than in the currently infected individuals ($p$ <0.001) and spontaneously eradicated cases ($p$ <0.001). We observed that each parameter overlapped in distribution among the groups, especially between the true uninfected and spontaneously eradicated groups, which made it difficult to judge infection status using only a single parameter (Table 1).

To better predict true uninfected individuals, we next performed a logistic regression analysis with the combination of several parameters. Statistically significant factors for true

**Table 1. Baseline data according to *Helicobacter pylori* infection status.**

| Variables | True uninfected (N = 528) | Currently infected (N = 97) | Spontaneously eradicated cases (N = 59) | True vs. Current | True Vs. Eradicated | Current vs Eradicated |
|---|---|---|---|---|---|---|
| Sex, male (%) | 250 (47.3) | 53 (54.6) | 22 (37.3) | | | |
| Age[†] | 54.0 (44.0–65.0) | 64.0(55.0–70.0) | 67.0 (59.0–71.5) | <0.001 * | <0.001 * | 0.094 |
| *H. pylori* antibody[†] | 1.5 (1.5–1.5) | 29.0 (14.0–100.0) | 3.0 (1.5–9.5) | <0.001 * | <0.001 * | <0.001 * |
| PGI[†] | 47.9 (40.0–59.0) | 68.3 (52.7–84.2) | 37.5 (26.0–52.9) | <0.001 * | <0.001 * | <0.001 * |
| PGII [†] | 6.7 (5.3–8.5) | 21.5 (15.6–27.1) | 6.4 (5.4–7.9) | <0.001 * | 0.858 | <0.001 * |
| PGI/II ratio[†] | 7.3 (6.4–8.3) | 3.3 (2.4–3.9) | 5.3 (4.0–7.2) | <0.001 * | <0.001 * | <0.001 * |

*$p$ <0.001,
[†]Median (IQR).

**Table 2. Variables associated with true uninfected status in logistic regression analysis.**

|  | Odds ratio | 95% CI | | p value |
|---|---|---|---|---|
| Sex | 0.634 | 0.329 | 1.223 | 0.174 |
| **Age** | 0.917 | 0.888 | 0.947 | **< 0.001**[**] |
| **HP antibody** | 0.663 | 0.584 | 0.751 | **< 0.001**[**] |
| **PGI** | 1.083 | 1.033 | 1.135 | **< 0.001**[**] |
| **PGII** | 0.725 | 0.564 | 0.933 | **0.012**[*] |
| PGI/II ratio | 1.018 | 0.685 | 1.513 | 0.930 |

[*] $p < 0.05$,

[**] $p < 0.001$.

uninfected status were age ($p < 0.001$), HP antibody ($p < 0.001$), PGI ($p < 0.001$), and PGII ($p = 0.012$) (Table 2).

The area under the curve (AUC), 95% CI, optimal cut-off value, sensitivity, and specificity for each parameter and their combinations are shown in Table 3. Since the outcome was a true uninfected status (true HP-uninfected: test positive, currently infected/spontaneously eradicated cases: test negative), we defined sensitivity, specificity, positive predictive value, and negative predictive value as follows: sensitivity: the percentage of HP-uninfected individuals who tested positive; specificity: the percentage of currently infected and spontaneously eradicated individuals who tested negative; positive predictive value (PPV): the probability that subjects with a positive test truly are HP-uninfected; and negative predictive value (NPV): the probability that subjects with a negative test truly are currently infected or spontaneously eradicated cases. The highest AUC was 0.944 with the combination of age, HP antibody, PGI, and PGII (+PGI/II ratio). Next, we tested the statistical significance of the difference between the AUCs of (A) HP antibody alone; (B) HP antibody, PGI, and PGI/II ratio; (C) age, HP antibody, PGI, and PGII (four-parameter model); and (D) age, sex, HP antibody, PGI, PGII, and PGI/II ratio (Fig 2). The results showed significant differences in the AUC of HP antibody alone (A) compared with combination models (B), (C), and (D), at $p = 0.008$, $p < 0.001$, and $p < 0.001$, respectively. In addition, the four-parameter model (C) had an AUC that was significantly different from that of the combination model with HP antibody, PGI, and PGI/II ratio (B) ($p = 0.023$). As logistic regression analysis using the four-parameter prediction formula showed the highest AUC for true uninfected status, we developed a formula (P) for predicting true uninfected

**Table 3. Results of logistic regression analysis for true uninfected individuals.**

|  | AUC | 95% CI | Cut-off value | Sensitivity (%) | Specificity (%) |
|---|---|---|---|---|---|
| Age (y/o) | 0.702 | 0.657–0.747 | 58 | 60.4 | 71.8 |
| HP antibody (U/ml) | 0.873 | 0.837–0.909 | <3.0 | 86.6 | 79.5 |
| PGI (ng/ml) | 0.559 | 0.497–0.622 | 58 | 73.7 | 50.6 |
| PGII (ng/ml) | 0.787 | 0.737–0.837 | 10.6 | 90.0 | 66.0 |
| PGI/II ratio | 0.891 | 0.853–0.930 | 5.7 | 89.0 | 83.3 |
| HP antibody, PGI/II ratio | 0.922 | 0.889–0.956 | 0.8371 | 89.8 | 85.9 |
| HP antibody, PGI/II ratio, PGI | 0.925 | 0.892–0.958 | 0.8456 | 89.4 | 88.5 |
| Age, HP antibody, PGI, PGII | 0.944 | 0.918–0.971 | 0.8135 | 93.2 | 88.5 |
| Age, HP antibody, PGI, PGII, PGI/II ratio | 0.944 | 0.917–0.971 | 0.8117 | 93.2 | 88.5 |
| Age, sex, HP antibody, PGI, PGII | 0.943 | 0.916–0.971 | 0.8383 | 91.3 | 89.7 |
| Age, sex, HP antibody, PGI, PGII, PGI/II ratio | 0.943 | 0.916–0.971 | 0.8375 | 91.3 | 89.7 |

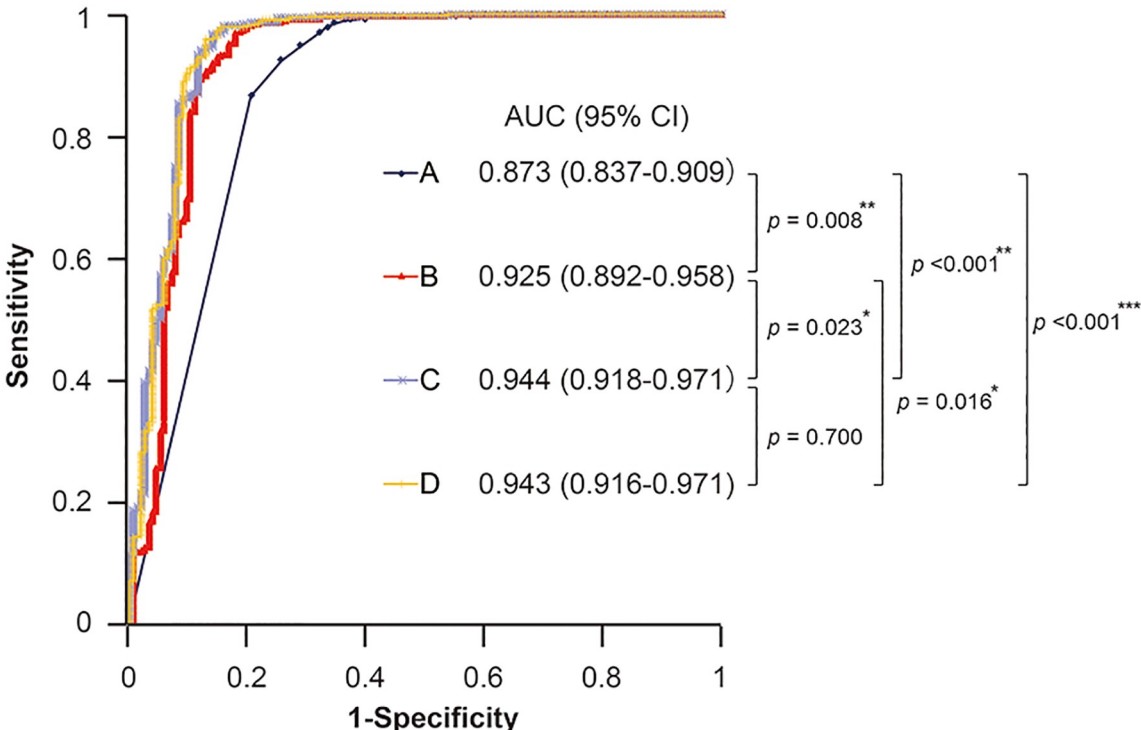

**Fig 2. Comparison of AUCs for true uninfected individuals among models with HP antibody and combined tests using PGs, age, and sex.** A. HP antibody. B. HP antibody, PGI, and PGI/II ratio. C. Age, HP antibody, PGI, and PGII (four-parameter model). D. Age, sex, HP antibody, PGI, PGII, and PGI/II ratio. *$p < 0.05$, **$p < 0.01$, ***$p < 0.001$.

status as follows: P = $1/(1+e^{-X})$, X = 7.0158−0.0869 (age)−0.4120 (HP antibody)+0.0784 (PGI)−0.3259 (PGII) (male = 1, female = 0). Because the optimal cut-off value using the ROC curve was calculated to be 0.8135, we defined the patients with P ≥0.8135 as true HP-uninfected individuals and those with P <0.8135 as currently infected or spontaneously eradicated individuals.

Finally, the sensitivity and specificity of the four-parameter combination (cut-off value 0.8135) were compared with those of the three-parameter model. The specificity was higher (88.5%) in the four-parameter prediction formula (cut-off value 0.8135) than in the three-parameter model using different HP antibody cut-off values of <3 U/ml (82.7%) and <10 U/ml (68.6%). In addition, the sensitivity of the four-parameter prediction formula (cut-off value 0.8135) was higher than that of the three-parameter model using the HP antibody cut-off value of <3 U/ml (93.2% vs. 86.6%). Next, we compared the diagnostic accuracy between cut-off values of 0.8135 and 0.90 because 0.90 was expected to have higher specificity (fewer cases with current or spontaneously eradicated infection but judged as being HP-uninfected by the prediction formula). Although the specificity of the cut-off value of 0.90 was 3.2% higher than that of 0.8135, the sensitivity of the former was 13.5% lower than that of the latter. In addition, although the PPV of the cut-off value of 0.90 was 0.5% higher than that of the cut-off value of 0.8135, the NPV of 0.90 was 22.1% lower than that of the cut-off value of 0.8135 (Table 4a). In summary, although we reduced the number of individuals falsely identified as being uninfected by five by shifting the cut-off value up from 0.8135 to 0.90, we misclassified 71 more true uninfected subjects. Therefore, we concluded that 0.8135 was the most appropriate cut-off value. As a result, our approach reduced both the number of individuals falsely identified as

**Table 4. Comparison of diagnostic accuracy between the traditional three-parameter model and the four-parameter prediction formula.**

| a | | Three-parameter model HP antibody(−) and PG>70 or PGII>3 | | Four-parameter prediction formula (the present study) | |
|---|---|---|---|---|---|
| | | HP antibody <3 U/ml | HP antibody <10 U/ml | Cut-off value 0.8135 | Cut-off value 0.9 |
| Specificity (%) | | 82.7 | 68.6 | 88.5 | 91.7 |
| Sensitivity (%) | | 86.6 | 99.1 | 93.2 | 79.7 |
| Positive predictive value (%) | | 94.4 | 91.4 | 96.5 | 97.0 |
| Negative predictive value (%) | | 64.5 | 95.5 | 79.3 | 57.2 |
| b | | Three-parameter model HP antibody(−) and PG>70 or PGII>3 | | Four-parameter prediction formula (the present study) | |
| | | HP antibody <3 U/ml | HP antibody <10 U/ml | Cut-off value 0.8135 | Cut-off value 0.9 |
| False uninfected subjects* (number) | | 27 | 49 | 18 | 13 |
| Misclassified true uninfected subjects** (number) | | 71 | 5 | 36 | 107 |

*False uninfected subjects: Currently infected and spontaneously eradicated subjects incorrectly classified into the true uninfected group.

**Misclassified true uninfected subjects: True uninfected subjects in the currently infected/spontaneously eradicated group.

uninfected, who should be classified as currently infected or spontaneously eradicated cases (from 27 to 18), and the number of true uninfected subjects misclassified into the currently infected/spontaneously eradicated group (from 71 to 36) (Table 4b).

## Discussion/Conclusion

This study was an observational study of healthy individuals who underwent medical checkups at our hospital. We selected sex, age, HP antibody, and PGs as parameters and developed a logistic regression formula to accurately predict true HP-uninfected status. Consequently, we determined that the prediction formula using age, HP antibody, PGI, and PGII was the best model. Our approach improved the sensitivity and specificity for true uninfected status compared with the traditional three-parameter model (HP antibody <3 U/ml and PGI >70 ng/ml or PGI/II ratio >3.0) (Table 4a). As a result, we reduced the number of currently infected and spontaneously eradicated individuals who were misclassified into the true uninfected group (Table 4b). Because the risk of gastric cancer differs markedly between true uninfected individuals and others (currently infected individuals and spontaneously eradicated cases), the prediction model might be useful for gastric cancer screening.

One of the key problems with the ABC classification system is that "false A" individuals, who are considered a high-risk group, fail to undergo endoscopy. Kishikawa et al. suggested using a PGI level of ≤37 ng/ml or a PGI/II ratio of ≤5.1 to identify spontaneously eradicated cases in Group A [4]. Similarly, Chinda et al. reported optimal cut-off values for PGI and the PGI/II ratio of ≤31.2 ng/ml and ≤4.6, respectively [4]. While these two research groups attempted to reduce the number of currently infected and spontaneously eradicated individuals in Group A, the present study aimed to diagnose "true uninfected individuals" specifically. As a result, we saw improvements in both the number of individuals falsely classified as uninfected (from 27 to 18) and the number of misclassified true uninfected subjects (from 71 to 36)

(Table 4b). Boda et al. also reported that a logistic regression analysis with multiple parameters, including sex, age, gastrin, and PGs, showed the best sensitivity and specificity for distinguishing high-risk patients from true HP-uninfected subjects. Although their target population was different from ours (gastric epithelial neoplasm patients vs. general healthy individuals during a medical checkup), logistic regression analysis with multiple parameters might be a promising approach for predicting true uninfected populations [10]. Since diagnostic accuracy was improved by using several parameters in our study and this previous one, adding other factors such as past history (ulcer), family history (gastric cancer), or social history (alcohol intake) that are thought to be associated with infection status might be useful to improve the prediction model [15, 16]. Meanwhile, we showed that age was an efficient predictor in the present study. However, a prediction model including age as a predictor should be reassessed in different study periods and geographical regions because the increasing prevalence of HP infection with age is not due to aging, but largely due to the environment during early childhood, such as the water supply system [17]. We should establish a prediction model by taking the prevalence of HP infection in the community into account.

There were several limitations to this study. First, since we used only SAT and endoscopic findings as an index for infection status, we could not identify false-negative SAT subjects. As SAT does not have 100% sensitivity and specificity [18, 19], several currently infected subjects with false-negative SAT might have been misclassified into the spontaneously eradicated group. In addition, we did not perform histological diagnosis, although it was sometimes difficult to judge HP-associated gastritis only by endoscopy. To establish a more accurate prediction model, adding other diagnostic tests (urea breath test, rapid urease test, and histology) might be required. Second, this study was based at a single center. A multicenter study is necessary to verify the results.

Our findings suggest that a prediction formula with the combination of noninvasive parameters of age, HP antibody, PGI, and PGII is useful for predicting true uninfected status without the need for endoscopic examination.

## Author Contributions

**Conceptualization:** Takako Takayama, Hideo Suzuki.

**Data curation:** Takako Takayama, Kazushi Maruo.

**Formal analysis:** Takako Takayama.

**Investigation:** Takako Takayama, Kazushi Maruo.

**Methodology:** Takako Takayama.

**Project administration:** Hideo Suzuki.

**Supervision:** Hideo Suzuki, Kosuke Okada, Takeshi Yamada, Kazushi Maruo, Yoko Saito, Yuji Mizokami.

**Writing – original draft:** Takako Takayama.

**Writing – review & editing:** Hideo Suzuki, Kosuke Okada, Takeshi Yamada, Kazushi Maruo, Yoko Saito, Yuji Mizokami.

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
