## [Decision Letter · Decision Letter 0]

24 Jul 2020

PONE-D-20-17227

Prediction of true Helicobacter pylori uninfected status by using a combination of Helicobacter pylori antibody, serum pepsinogen, and other factors: Logistic regression analysis

PLOS ONE

Dear Dr. Suzuki,

Thank you for submitting your manuscript to PLOS ONE. After careful consideration, we feel that it has merit but does not fully meet PLOS ONE’s publication criteria as it currently stands. Therefore, we invite you to submit a revised version of the manuscript that addresses the points raised during the review process.

Prediction of a condition without invasive procedures is always interesting. I myself have also worked with that problem years back (*Borch, K., Axelsson, C. K., Halgreen, H., Nielsen, M. D., Ledin, T., & Szesci, P. B. (1989). The ratio of pepsinogen A to pepsinogen C: a sensitive test for atrophic gastritis. Scandinavian journal of gastroenterology, 24(7), 870-876; Burman, P., Karlsson, F. A., Lööf, L., Szecsi, P. B., & Borch, K. (1991). H+ K+-ATPase antibodies in autoimmune gastritis: observations on the development of pernicious anemia. Scandinavian journal of gastroenterology, 26(2), 207-214*) however gastroscopy is still the choice.

A predictive model has rather stick demands, especially with sensitivity if invasive procedure could be avoided. I would like a discussion on negative predictive value in light of the relative low incidence of the disease, even in higher risk areas.

We look forward to receiving your revised manuscript.

Kind regards,

Pal Bela Szecsi, M.D. D.M.Sci.

Academic Editor

PLOS ONE

Journal Requirements:

2. Thank you for stating in the text of your manuscript " The study was approved by the University Hospital of Tsukuba Ethics Committee (H29-304)." Please also add this information to your ethics statement in the online submission form.

3. In your ethics statement in the Methods section and in the online submission form, please provide additional information about the data used in your retrospective study. Specifically, please ensure that you have discussed whether all data were fully anonymized before you accessed them and/or whether the IRB or ethics committee waived the requirement for informed consent. If patients provided informed written consent to have data from their medical records used in research, please include this information.

Reviewers' comments:

Reviewer's Responses to Questions

**Comments to the Author**

1. Is the manuscript technically sound, and do the data support the conclusions?

Reviewer #1: Partly

Reviewer #2: Partly

2. Has the statistical analysis been performed appropriately and rigorously? 

Reviewer #1: No

Reviewer #2: Yes

3. Have the authors made all data underlying the findings in their manuscript fully available?

Reviewer #1: No

Reviewer #2: Yes

4. Is the manuscript presented in an intelligible fashion and written in standard English?

Reviewer #1: Yes

Reviewer #2: Yes

5. Review Comments to the Author

Reviewer #1: This is an interesting study which to discriminate uninfected individuals from the subjects with past or current Hp infection “without endoscopic examination”. These are suggestions to improve the manuscript.

Major comments:

1. Main flaw of this study is the discrepancy found between the results and conclusions. Although uninfected subjects were younger in age, showed negative Hp serology test finding, higher serum PG I concentration, and lower PG II concentration, there is no combination of these four significant variables even in Figure 2. Because the authors stick too much to all 6 variables, it was concluded that a combination of 6 noninvasive parameters (sex, age, HP antibody, PGI, PGII and the PGI/II ratio). is useful in predicting true HP uninfected individuals without endoscopic examination. Please do not stick on comparing between the accuracies of 6 variables and 3 variables (HP antibody, PGI and PGI/II ratio).

2. Because the “true uninfected individuals” were determined by a negative stool antigen test and the lack of advanced CAG (> closed-type 1) or MG on endoscopic finding, there might be a discrepancy with the results derived by using 6 noninvasive parameters (sex, age, HP antibody, PGI, PGII and the PGI/II ratio). For clarification, the cutoff values of age, serum PG I level, and PG II level should be shown with their accuracies.

3. For better clinical application of the prediction formula, 467 subjects with past eradication should be tested as well.

Minor comments:

1. Rewrite the Methods and Results under the subheading, and please do not insert the tables and figures between the main body. Put them at the end of the manuscript.

2. The title looks less scientific because of “other factors”.

Reviewer #2: The authors developed accurate HP prediction model. The prediction model had higher sensitivity and specificity than conventional model. However, The authors did not explain the meaning of HP prediction. Accurate HP prediction model cause effective screening. The authors must discuss the usefulness of the study.

1. The purpose of ABCD classification is not only prediction of HP infection status, but also evaluation of gastric cancer risk. Risk stratification by ABCD classification might enable personalized prevention of gastric cancer. The new HP status prediction might lead better gastric cancer screening, however, nothing was described concerning gastric cancer prevention.

2. Age was one of the significant variable to predict HP infection in the study. However, some evidences suggested birth cohort (birth year) rather than age was truly associated with HP infection. I am interested in the model including birth cohort as a predictor.

3. Sex was included in the 6-parameter prediction formula, although it did not have significant impact on HP infection status. Did it improve the discriminatory ability of the model?

4. Describe a definition of sensitivity and specificity in the study. In usual case, sensitivity means a proportion of test positives among diseased subjects and specificity means a proportion of test negatives among healthy subjects. Although no definition of sensitivity and specificity in the study, I guessed a proportion of uninfected defined by test (true uninfected) among all uninfected subjects was sensitivity. Thus, the definition of sensitivity and specificity were opposite to the usual ones. It would confuse readers to assess the results.

5. The study design aimed to specify true uninfected population. Why the authors specify uninfected population by the prediction model? Some intervention would be necessary for HP infected group to prevent gastric cancer, while uninfected people do not need any help. Therefore, it is natural to find HP infected (current HP infection + spontaneously eradicated) by the prediction model.

6. Suppose the prediction model is used for gastric cancer prevention, high risk population (HP infected population) should be detected for effective screenings. Examination with high sensitivity was relevant for screening despite of relatively less specificity, because of less false negatives. As for gastric cancer screening, minimum number of cases who were HP infected but judged as HP uninfected by examinations, should be required. Were the cut-off value of 91.3% sensitivity and 89.7% specificity appropriate?

6. PLOS authors have the option to publish the peer review history of their article (what does this mean?). If published, this will include your full peer review and any attached files.

Reviewer #1: No

Reviewer #2: No

---

## [Author Response · Author response to Decision Letter 0]

30 Aug 2020

Please find "response to reviewer" file. All the responses were mentioned in the file.

---

## [Decision Letter · Decision Letter 1]

18 Sep 2020

Prediction of true Helicobacter pylori-uninfected status using a combination of age, serum antibody and pepsinogen: Logistic regression analysis

PONE-D-20-17227R1

Dear Dr. Suzuki,

We’re pleased to inform you that your manuscript has been judged scientifically suitable for publication and will be formally accepted for publication once it meets all outstanding technical requirements.

Kind regards,

Pal Bela Szecsi, M.D. D.M.Sci.

Academic Editor

PLOS ONE

Additional Editor Comments (optional):

Reviewers' comments:

Reviewer's Responses to Questions

**Comments to the Author**

1. If the authors have adequately addressed your comments raised in a previous round of review and you feel that this manuscript is now acceptable for publication, you may indicate that here to bypass the “Comments to the Author” section, enter your conflict of interest statement in the “Confidential to Editor” section, and submit your "Accept" recommendation.

Reviewer #1: All comments have been addressed

Reviewer #2: All comments have been addressed

2. Is the manuscript technically sound, and do the data support the conclusions?

Reviewer #1: Yes

Reviewer #2: (No Response)

3. Has the statistical analysis been performed appropriately and rigorously? 

Reviewer #1: Yes

Reviewer #2: (No Response)

4. Have the authors made all data underlying the findings in their manuscript fully available?

Reviewer #1: Yes

Reviewer #2: (No Response)

5. Is the manuscript presented in an intelligible fashion and written in standard English?

Reviewer #1: No

Reviewer #2: (No Response)

6. Review Comments to the Author

Reviewer #1: Thank you for answering to my queries. Methods and Results are still written without subtitles, but I have nothing to mention further if it matches the journal format.

Reviewer #2: (No Response)

7. PLOS authors have the option to publish the peer review history of their article (what does this mean?). If published, this will include your full peer review and any attached files.

Reviewer #1: No

Reviewer #2: No

---

## [Editor Report · Acceptance letter]

22 Sep 2020

PONE-D-20-17227R1

Prediction of true *Helicobacter pylori*-uninfected status using a combination of age, serum antibody and pepsinogen: Logistic regression analysis

Dear Dr. Suzuki:

I'm pleased to inform you that your manuscript has been deemed suitable for publication in PLOS ONE. Congratulations! Your manuscript is now with our production department.

Kind regards,

on behalf of

Dr. Pal Bela Szecsi 

Academic Editor

PLOS ONE